# CROSS-PROBE BERT FOR EFFICIENT AND EFFECTIVE CROSS-MODAL SEARCH

## ABSTRACT

Inspired by the great success of BERT in NLP tasks, many text-vision BERT models emerged recently. Benefited from cross-modal attentions, text-vision BERT models have achieved excellent performance in many language-vision tasks including text-image retrieval. Nevertheless, cross-modal attentions used in text-vision BERT models require too expensive computation cost when solving text-vision retrieval, which is impractical for large-scale search. In this work, we develop a novel architecture, cross-probe BERT. It relies on devised text and vision probes, and the cross-modal attentions are conducted on text and vision probes. It takes lightweight computation cost, and meanwhile effectively exploits cross-modal attention. Systematic experiments conducted on two public benchmarks demonstrate state-of-the-art effectiveness and efficiency of the proposed method.

## 1 INTRODUCTION

Tradition text-to-image retrieval tasks are tackled by joint embedding methods. They map the text queries and reference images into the same feature space so that queries and images can be compared directly. Basically, they adopt a two-tower architecture, in which one tower extracts features of text queries, and the other tower extracts features of reference images. In the training phase, the parameters of two towers are optimized so that a query and its relevant images are close in the feature space whereas the distance between the query and its irrelevant images is large. Since two towers independently generate image and query features, image features can be extracted offline and cached in the database. In the search phase, the cached image features can be directly compared with the query's feature, and the retrieval is efficient. Due to high efficiency, joint embedding methods based on the two-tower structure have been widely used in many large-scale cross-modal retrieval.

Inspired by great success achieved by self-attention mechanism of Transformer (Vaswani et al. (2017)) and BERT (Devlin et al. (2019)) in NLP tasks, several text-vision BERT models (Lu et al. (2019); Li et al. (2020)) emerge. They take the query-image pair as input and extend the original text-modal self-attention to the multi-modal self-attention. The text-vision BERT effectively models the interactions between image features and query features, provides contextual encoding for both image features and query features, and achieves a significantly better retrieval accuracy compared with its two-tower counterpart. Despite the high effectiveness achieved by text-vision BERT, the extremely high computation cost brought by pairwise input limits its practical usefulness, especially for large-scale cross-modal retrieval in industrial applications. Given a query and $N$ reference images, it needs to feed $N$ query-image pairs to text-vision BERT for $N$ relevance scores. That is, it requires to repeatedly encode the query for $N$ times. In a large-scale cross-modal retrieval task, $N$ is extremely large, making text-vision BERT prohibitively slow for obtaining relevant scores with all reference images. In contrast, a two-tower encoder only needs to encode the query for one time, and $N$ reference image features can be pre-computed offline and cached in the database. Thus, it obtains relevance scores between the query and reference images in a very efficient manner by computing the cosine similarities between the query feature and pre-computed reference image features.

Though the inefficient pairwise attention limits the usefulness of the text-vision BERT in large-scale cross-modal retrieval, there are few works to speed up the text-vision BERT. In fact, the inefficiency caused by pairwise input is a general problem which is also encountered in other retrieval tasks such as query-to-document retrieval (Humeau et al. (2020)), question answering (Cao et al. (2020); Zhang et al. (2020)). In these tasks, similarly there are two mainstream encoders for obtaining the relevance score. The first type of encoder, Bi-encoder (Dinan et al. (2019); Mazare et al. (2018)), is based on

the two-tower architecture. Since the query/question and document are independently encoded, the document features can be pre-computed and cached. In this case, the relevance between the query and each document can be determined by the cosine similarity between the query/question's feature and the document's cached feature. It achieves a high efficiency but a relatively low retrieval accuracy. In contrast, Cross-encoder (Urbanek et al.) takes a question-answer pair or a query-document pair as input, exploiting the cross-attention like text-vision BERT and achieving high retrieval accuracy but is inefficient. To balance effectiveness and efficiency, existing methods (Cao et al. (2020); Zhang et al. (2020)) adopt the two-tower architecture in lower layers and use the cross-attention architecture in the upper layers. We term this architecture as "split-merge" encoder. In that case, features from lower two-tower layers can be extracted offline and cached. Then question-answer or query-document attentions can be conducted in the upper layers. Since the number of upper cross-attention layers is small, the efficiency is boosted. Similarly, Poly-Encoder (Humeau et al. (2020)) conducts the two-tower architecture for feature extraction, and uses an additional cross-attention layer on the top to obtain the similarities between the query and reference items.

In this paper, we propose a novel architecture, cross-probe (CP) BERT for effective and efficient cross-modal retrieval. Motivated by the great success of the "split-merge" style encoder in query document retrieval, we extend text-vision BERT to adopt it for speeding up the computation. In particular, we devise several vision probes and text probes along with the image's local features and the query's word features. In the lower a few layers, we adopt the two-tower architecture. The vision probes and the image's local features are concatenated and fed into the vision tower and generate the attended vision probes. In parallel, the text probes and the query's word features are concatenated and fed into the text tower, and generates the attended text probes. After that, the attended vision probes and text probes are concatenated and fed into a series of cross-attention layers to exploit the cross-modal attentions. Since the number of text probes is considerably smaller than the number of words in the query and the number of vision probes is smaller than the number of local features of the image, the cost of our CP BERT in computing cross-modal attention is significantly less than that of text-vision BERT. Meanwhile, the cross-modal attention is only exploited in the upper a few layers, making our CP BERT more efficient. Systematic experiments conducted on two public benchmarks demonstrate the excellent effectiveness and efficiency of the proposed CP BERT.

## 2 RELATED WORK

Traditional cross-modal retrieval, *e.g.*, text-image retrieval, relies on joint embedding. It maps texts and images from two modalities into a common feature space through two encoders. Then texts and images can be compared and the distance between their global features in the common feature space measures their similarities. Early joint embedding methods (Gong et al. (2012); Rasiwasia et al. (2010)) utilize canonical correlation analysis (CCA) to project hand-crafted text and image features to a joint CCA space. Recently, inspired by great progress achieved by deep neural network, methods based on deep learning emerge. VSE++ (Faghri et al. (2017)) obtains the image feature through a convolution neural network (CNN) and encodes the text by a gated recurrent unit (GRU). The CNN and the GRU are trained in an end-to-end manner by the designed triplet loss. The triplet loss seeks to minimize the distance between the relevance text-image pairs and maximize the distance between irrelevant pairs. The merit of joint-embedding methods is simplicity and efficiency. In this case, a text query as well as a image is represented by a global feature. The relevance between the image and the text can be efficiently obtained by computing the cosine similarity of their features. Meanwhile, the global text and image features make it feasible for the approximate nearest neighborhood (ANN) search such as Hashing and inverted indexing, so that the large-scale retrieval can be efficient.

Nevertheless, the global feature used in joint-embedding methods has limitations. In many cases, the relevance between a text and an image is determined by very few words in the text and some small regions in the image. Therefore, the relevant text words and image regions might be distracted by irrelevant words and regions when using global features. Thus methods based on local features are proposed to overcome the limitations of global features used in joint-embedding methods. In DVS (Karpathy & Fei-Fei (2014)), an image is represented by a set of bounding box features extracted from the object detector, R-CNN. Meanwhile, the text is represented by a sequence of word features extracted from an RNN. Then the bounding box features and word features are aligned to obtain the similarity between the image and the text. The alignment operation can effectively alleviate the distraction from irrelevant word-region pairs. Similarly, SCAN (Lee et al. (2018)) relies on bounding box features from a faster R-CNN and word features from a GRU. It conducts the alignment through soft-attention and optimizes the loss function through hard negative mining. Nevertheless, both DVS

and SCAN conduct the alignment in the late stage, which limits their effectiveness due to a lack of thorough interactions between bounding box features of the image and word features of the query.

To conduct the interactions between bounding box features and word feature thoroughly, text-vision BERT methods are proposed. They can be categorized into two types: one-stream (Li et al. (2019; 2020); Chen et al. (2020)) structure and two-stream (Lu et al. (2019); Tan & Bansal (2019)) structure. A one-stream structure concatenates word features and bounding box features into a long sequence, and conducts a series of self-attention operations for cross-modal interactions. In parallel, a two-stream structure designs two streams for word features and bounding box features, respectively. In the text stream, word features are attended by bounding box features, and in the vision stream, bounding box features are attended by word features. Benefited from thorough interactions between bounding box features and word features, text-vision BERT achieves state-of-the-art performance in many vision-language tasks. Nevertheless, text-vision BERT requires extremely expensive computational cost, limiting its usefulness in large-scale cross-modal retrieval applications.

## 3 METHOD

Following existing text-vision BERT models (Lu et al. (2019); Li et al. (2020)), for each reference image $I$, a set of bounding box features is extracted through faster R-CNN (Ren et al. (2015)) pre-trained on Visual Genome (Krishna et al. (2017)). The set of bounding box features, $\mathcal{B} = \{\mathbf{b}_i\}_{i=1}^{N}$ is the initial representation of $I$. In parallel, the query sentence $q$, goes through a word-embedding layer to generate word features, $\mathcal{W} = \{\mathbf{w}_i\}_{i=1}^{M}$, which is the initial representation of $q$. The ranking model (RM) takes $\mathcal{B}$ and $\mathcal{W}$ as input, and generates the similarity score between $q$ and $I$:

$$s(q, I) = \mathrm{RM}(\mathcal{B}, \mathcal{W}).$$

In the search phase, the similarity scores $s(q, I_1), \cdots, s(q, I_N)$ are used for ranking images. In the training phase, the similarity scores are used for constructing the triplet loss with the margin $m$:

$$\mathcal{L} = [s(q, I_-) - s(q, I_+) + m]_+.$$

Below, we introduce the architectures of existing ranking models and propose our cross-probe BERT.

### 3.1 TWO-TOWER BERT

The two-tower BERT ($\mathrm{BERT_{TT}}$), visualized in Figure 1(a), models the image and the query separately, generates the query's feature $\hat{\mathcal{W}}$ and the reference image's feature $\hat{\mathcal{B}}$ through two BERTs:

$$\bar{\mathcal{W}} = \mathrm{BERT_T}(\mathcal{W}), \quad \bar{\mathcal{B}} = \mathrm{BERT_V}(\mathcal{B}).$$

The text query feature $\bar{\mathbf{q}}$ is set as $\bar{\mathbf{w}}_1$, the attended feature of the first token in $\bar{\mathcal{W}}$. The image feature $\bar{\mathbf{I}}$ is obtained by sum-pooling the attended bounding box features in $\bar{\mathcal{B}}$ by $\bar{\mathbf{I}} = \sum_{\bar{\mathbf{b}} \in \bar{\mathcal{B}}} \bar{\mathbf{b}}$. The similarity score between the query and the image is obtained by the cosine similarity between $\bar{\mathbf{I}}$ and $\bar{\mathbf{q}}$. In two-tower BERT model, the query feature is independent with reference image features. This independent property makes two-tower BERT model quite suitable for large-scale cross-modal retrieval. Let us denote the computation cost of $\mathrm{BERT_T}$ per query by $c_T$ and denote that of $\mathrm{BERT_V}$ per reference image by $c_V$. Given a query and $K_I$ reference images, the total complexity cost in feature extraction to obtain their similarity scores is $C_{TT} = c_T + K_I c_V$. Since features of $K_I$ reference images can be pre-computed in the offline phase, the factual complexity cost in the online search phase, $\hat{C}_{TT}$, is only taken on the query-side feature extraction:

$$\hat{C}_{TT} = c_T.$$

Later, we will show that the online computational cost of $\mathrm{BERT_{TT}}$, $\hat{C}_{TT}$, is significantly lower than that used in text-vision BERT, making it widely used in large-scale cross-modal retrieval.

### 3.2 TEXT-VISION BERT

Existing text-vision BERT ($\mathrm{BERT_{TV}}$) models can be categorized into two groups: single-stream architecture and two-stream architecture. Since their performance is comparable, we only introduce the single-stream architecture here. $\mathrm{BERT_{TV}}$, visualized in Figure 1 (b), concatenates the initial word features of the query and the initial bounding box features of the reference image as the input,

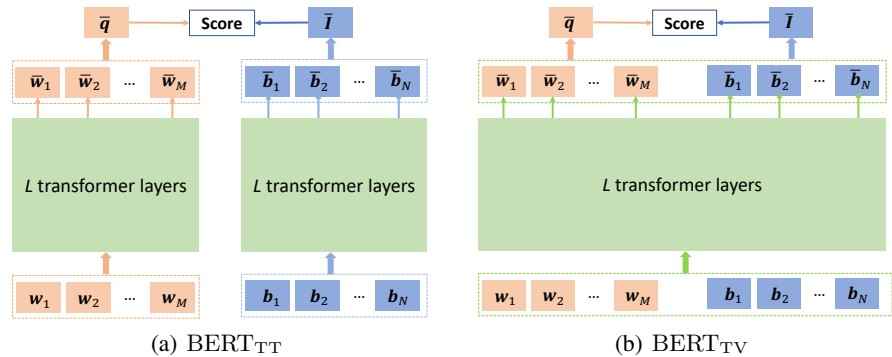

Figure 1: Architecture of two-tower BERT (BERT$_{\text{TT}}$) and vision-text BERT (BERT$_{\text{TV}}$).

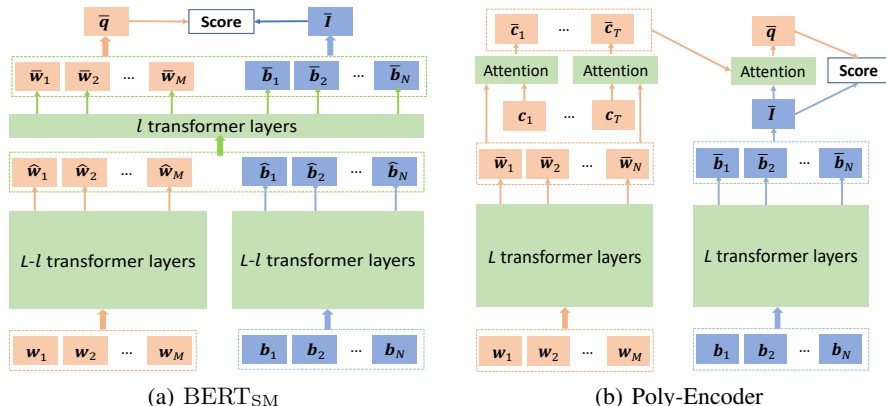

Figure 2: Architecture of split-merge BERT (BERT$_{\text{SM}}$) and Poly-Encoder.

$\mathcal{M} = [\mathcal{W}, \mathcal{B}]$. Then the attended word features and attended bounding box features based on cross-modal attentions are obtained through BERT$_{\text{TV}}$ consisting of a series of Transformer blocks:

$$[\hat{\mathcal{W}}, \hat{\mathcal{B}}] = \text{BERT}_{\text{TV}}([\mathcal{W}, \mathcal{B}]).$$

Then the global feature of the query $q$ and that of the image $I$ are obtained in the same manner as BERT$_{\text{TT}}$. The relevance between $q$ and $I$ is the cosine similarity between their global features.

Different from BERT$_{\text{TT}}$ which only exploits the intra-modal attentions, BERT$_{\text{TV}}$ additionally exploits the cross-modal attentions. At this point, the image feature is adaptively tuned based on the query, and the query feature is refined by the image as well. Benefited from the query-adapt image features and image-adapt query features brought by cross-modal attentions, BERT$_{\text{TV}}$ achieves a significantly higher retrieval accuracy than BERT$_{\text{TT}}$. On other hand, the query-adapt and image-adpat mechanism requires to recompute the reference image's representation for different queries and meanwhile need recompute the query's representation for different images, making BERT$_{\text{TV}}$ much more computationally expensive than its counterpart, BERT$_{\text{TT}}$. To be specific, given a query $q$ and $K_{\text{I}}$ images, to obtain their similarities, we need feed $K_{\text{I}}$ text-image pairs to BERT$_{\text{TV}}$. Let us denote the computation cost per text-image pair in BERT$_{\text{TV}}$ by $c_{\text{TV}}$, the total computation cost of feature extraction to obtain the similarities between the query $q$ and $K_{\text{I}}$ images is

$$C_{\text{TV}} = K_{\text{I}} c_{\text{TV}}$$

Due to pairwise input, the whole feature extraction should be done in the online search phase, that is, the online cost $\hat{C}_{\text{TV}} = C_{\text{TV}}$. Note that, in large-scale cross-modal retrieval task, $K_{\text{I}}$ is large, the online computation cost $\hat{C}_{\text{TV}}$ will be prohibitively large for real-time retrieval.

### 3.3 SPLIT-MERGE BERT

To boost the efficiency, Deformer (Cao et al. (2020)) and DC-BERT (Zhang et al. (2020)) decouple the query and reference items in lower layers and conduct the cross attention only in upper layers.

We term this structure as split-merge BERT ($\text{BERT}_{\text{SM}}$). We define the total number of transformer layers of $\text{BERT}_{\text{SM}}$ as $L$. Among them, the upper $l$ transformer layers adopt the architecture used in $\text{BERT}_{\text{TV}}$ and the lower $L - l$ layers adopt the two-tower architecture. Given a query sentence $q$ and $K_{\text{I}}$ images, to obtain their similarities, the total computation cost of feature extraction is

$$C_{\text{SM}} = \frac{L-l}{L}c_{\text{T}} + \frac{L-l}{L}K_{\text{I}}c_{\text{V}} + \frac{l}{L}K_{\text{I}}c_{\text{TV}}.$$

Since image features from two-tower layers can be pre-computed and cached, the online cost is:

$$\hat{C}_{\text{SM}} = \frac{L-l}{L}c_{\text{T}} + \frac{l}{L}K_{\text{I}}c_{\text{TV}}.$$

When $l = 0$, $\text{BERT}_{\text{SM}}$ degenerates to $\text{BERT}_{\text{TT}}$, which enjoys high efficiency but lacks cross-modal attentions. On the other hand, when $l = L$, it degenerates to $\text{BERT}_{\text{TV}}$ which exploits cross-modal attentions but takes high computation cost. The speedup of $\text{BERT}_{\text{SM}}$ over $\text{BERT}_{\text{TV}}$ is

$$\alpha_{\text{SM}} = \frac{\hat{C}_{\text{TV}}}{\hat{C}_{\text{SM}}} = \frac{K_{\text{I}}c_{\text{TV}}}{\frac{L-l}{L}c_{\text{T}} + \frac{l}{L}K_{\text{I}}c_{\text{TV}}}.$$

Since $K_{\text{I}}c_{\text{TV}} \gg c_{\text{T}}$, $\alpha_{\text{SM}}$ is around $L/l$. Thus, to achieve a high speed-up ratio, $l$ should be small. But a small $l$ might not fully exploit the cross-modal attentions, limiting its effectiveness for high-accuracy cross-modal retrieval. We visualize the architecture of split-merge BERT in Figure 2 (a).

### 3.4 POLY-ENCODER

Poly-Encoder (Humeau et al. (2020)) is another architecture for fast self-attention operation over pairs. Originally, it is used for sentence-document retrieval. We can also use it for cross-modal retrieval. To be specific, Poly-Encoder adopts the two-tower architecture to generate word features of the query and bounding boxes features of the reference image. On the image side, the bounding boxes features are pooled into a global reference image feature $\hat{\mathbf{I}}$. On the text side, it uses $m$ context codes to attend over the word features from text BERT to generate $m$ attended word features. Then these $m$ attended word features are further attended by $\hat{\mathbf{I}}$ to generate the global feature of the text query $\hat{\mathbf{q}}$. Then the similarity between the reference image and the query sentence is determined by the cosine similarity between $\hat{\mathbf{I}}$ and $\hat{\mathbf{q}}$. Since text-vision attentions are only conducted on the last layer, Poly-Encoder might not effectively models the interactions between vision features and text features. We visualize the architecture of Poly-Encoder in Figure 2 (b).

### 3.5 CROSS-PROBE BERT

We propose a cross-probe BERT architecture for fast cross-modal retrieval, and meanwhile exploits the text-vision attentions. For the easiness of illustration, we divide the cross-probe BERT into upper part and lower part. The lower part adopt a two-tower architecture. As shown in Figure 3, we devise $n$ vision probes $\mathcal{P}^v = \{\mathbf{p}_i^v\}_{i=1}^n$ and feed them along with $N$ bounding boxes features $\mathcal{B} = \{\mathbf{b}_i\}_{i=1}^N$ into a vision-stream BERT. The probes are learned weights, which are randomly initialized. Utilizing the lower $L - l$ transformer layers of the vision-stream BERT, the image-attend vision probes $\hat{\mathcal{P}}^v$ and bounding box features $\hat{\mathcal{B}}$ are obtained:

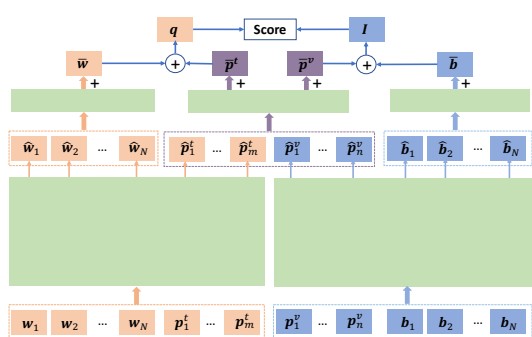

Figure 3: The architecture of cross-probe BERT.

$$[\hat{\mathcal{P}}^v, \hat{\mathcal{B}}] = \text{BERT}_{\text{V}}^{[1:L-l]}([\mathcal{P}^v, \mathcal{B}]).$$

In parallel, we devise $m$ text probes $\mathcal{P}^t = \{\mathbf{p}_i^t\}_{i=1}^m$. $\mathcal{P}^t$ and word features $\mathcal{W} = \{\mathbf{w}_i\}_{i=1}^M$ are fed into the text-stream BERT. Using the lower $L - l$ transformer layers of the text-stream BERT, query-attend text probes $\hat{\mathcal{P}}^t$ and self-attend word features $\hat{\mathcal{W}}$ are generated:

$$[\hat{\mathcal{P}}^t, \hat{\mathcal{W}}] = \text{BERT}_{\text{T}}^{[1:L-l]}([\mathcal{P}^t, \mathcal{W}]).$$

In the upper part, query-attend text probes $\hat{\mathcal{P}}^t$ and image-attend vision probes $\hat{\mathcal{P}}^v$ are concatenated, go through a text-vision BERT with $l$ self-attention layers and obtain cross-modal attended features:

$$[\bar{\mathcal{P}}^v, \bar{\mathcal{P}}^t] = \text{BERT}_{\text{C}}^{[1:l]}([\hat{\mathcal{P}}^t, \hat{\mathcal{P}}^v]).$$

Then sum-pooling is conducted on cross-modal attended probes $\bar{\mathcal{P}}^v$ and $\bar{\mathcal{P}}^t$ to obtain global features:

$$\bar{\mathbf{p}}^v = \sum_{\mathbf{p} \in \bar{\mathcal{P}}^v} \mathbf{p}, \quad \bar{\mathbf{p}}^t = \sum_{\mathbf{p} \in \bar{\mathcal{P}}^t} \mathbf{p}.$$

In parallel, the self-attended word features $\hat{\mathcal{W}}$ from the lower $L - l$ layers of the text-stream BERT are processed by the upper $l$ layers of the text-stream BERT. In a similar manner, the self-attended bounding box features $\hat{\mathcal{B}}$ are processed by the upper $l$ layers of the vision-stream BERT:

$$\bar{\mathcal{W}} = \text{BERT}_{\text{T}}^{[L-l+1:L]}(\hat{\mathcal{W}}), \quad \bar{\mathcal{B}} = \text{BERT}_{\text{V}}^{[L-l+1:L]}(\hat{\mathcal{B}}).$$

$\bar{\mathcal{W}}$ and $\bar{\mathcal{B}}$ are further sum-pooled into the global text feature and the global image feature:

$$\bar{\mathbf{w}} = \sum_{\mathbf{w} \in \bar{\mathcal{W}}} \mathbf{w}, \quad \bar{\mathbf{b}} = \sum_{\mathbf{b} \in \bar{\mathcal{B}}} \mathbf{b}.$$

Finally, we obtain the query's global feature $\mathbf{q}$ and the reference image's global feature $\mathbf{I}$ by

$$\mathbf{q} = \bar{\mathbf{w}} + \bar{\mathbf{p}}^t, \quad \mathbf{I} = \bar{\mathbf{b}} + \bar{\mathbf{p}}^v.$$

The similarity score between the reference image and the query text is calculated by the cosine similarity between $\mathbf{q}$ and $\mathbf{I}$. Given a query sentence $q$ and $K_{\text{I}}$ reference images, to obtain their similarity scores, we need compute once $L$-layer $\text{BERT}_{\text{T}}$, $K_{\text{I}}$ times $L$-layer $\text{BERT}_{\text{V}}$ and $K_{\text{I}}$ times $l$-layer $\text{BERT}_{\text{C}}$. We denote the computational cost per query of $\text{BERT}_{\text{T}}$ by $c_{\text{T}}$, the computational cost per image of $\text{BERT}_{\text{V}}$ by $c_{\text{V}}$, and the computational cost per query-image pair of $\text{BERT}_{\text{C}}$ by $c_{\text{C}}$, thus the total computational cost of the proposed cross-probe BERT for feature extraction is

$$C_{\text{CP}} = c_{\text{T}} + K_{\text{I}}c_{\text{V}} + K_{\text{I}}c_{\text{C}}.$$

Since features from vision-stream BERT can be pre-computed, the online cost of the retrieval is only

$$\hat{C}_{\text{CP}} = c_{\text{T}} + K_{\text{I}}c_{\text{C}}.$$

Note that, since $\text{BERT}_{\text{C}}$ consists of $l$ layers, and the length of input of $\text{BERT}_{\text{C}}$ is only $m + n$, the cost per pair of $\text{BERT}_{\text{C}}$ is significantly smaller than that of $\text{BERT}_{\text{TV}}$. Let us denote the input length of a transformer layer as $T$, then the complexity of the self-attention operation is in linear with $T^2$, whereas the feed-forward operation is in linear with $T$. The complexity of a transformer layer is

$$c(T) = \alpha T + \beta T^2,$$

where $\alpha$ is a unit cost for feed-forward operation and $\beta$ is a unit cost for self-attention operation. In this case, the speed-up ratio of the proposed cross-probe BERT over the text-vision BERT is

$$\alpha_{\text{CP}} = \frac{\hat{C}_{\text{TV}}}{\hat{C}_{\text{CP}}} = \frac{K_{\text{I}}L(\alpha(M+N) + \beta(M+N)^2)}{c_{\text{T}} + K_{\text{I}}l(\alpha(m+n) + \beta(m+n)^2)}.$$

## 4 EXPERIMENTS

We evaluate the proposed model on two public benchmarks for the cross-modal retrieval task, MS-COCO (Lin et al. (2014)) and Flickr30K (Young et al. (2014)) datasets. MS-COCO dataset consists of $123,287$ images, and each image is paired with five text descriptions. It was originally split into $82,783$ training images, $5,000$ validation images and $5,000$ testing images. Following (Karpathy & Fei-Fei (2014)), we add $30,504$ images that were originally in the validation set of MS-COCO to the training set. Meanwhile, we adopt 1K testing settings on MS-COCO dataset. Flickr30K contains $31,783$ images collected from the Flickr website. Following (Karpathy & Fei-Fei (2014)), we split the dataset into $29,783$ training images, $1,000$ validation images and $1,000$ testing images. We evaluate the cross-modal retrieval performance through image-to-text and text-to-image recall@K, which is the percentage of ground-truth matchings appearing in the top K-ranked results. Though recall@1 is the most widely used metric in real-world applications, we also report recall@$\{5, 10\}$.

| | MS-COCO | | | | | | Flickr30K | | | | | |
| | query2image R@ | | | image2query R@ | | | query2image R@ | | | image2query R@ | | |
| | 1 | 5 | 10 | 1 | 5 | 10 | 1 | 5 | 10 | 1 | 5 | 10 |
|---|---|---|---|---|---|---|---|---|---|---|---|---|
| TT | 58.3 | 86.5 | 91.8 | 68.3 | 93.5 | 97.4 | 42.5 | 71.4 | 79.6 | 54.9 | 81.0 | 88.2 |
| PE | 59.4 | 88.0 | 93.5 | 71.4 | 93.9 | 97.9 | 45.9 | 74.8 | 82.6 | 56.1 | 82.1 | 89.1 |
| SM | 60.6 | 89.6 | 95.3 | 74.2 | 94.4 | 97.6 | 45.1 | 74.8 | 84.2 | 57.6 | 82.2 | 89.9 |
| TV | 66.3 | 91.7 | 96.3 | 80.6 | 96.4 | 98.7 | 55.0 | 83.3 | 89.3 | 71.2 | 91.3 | 96.5 |
| CP | 66.0 | 91.0 | 95.3 | 80.2 | 96.1 | 98.2 | 55.0 | 80.3 | 86.1 | 70.8 | 89.6 | 94.1 |

Table 1: Comparisons among two-tower (TT) BERT, text-vision (TV) BERT, poly-encoder (PE), split-merge (SM) BERT and our cross-probe (CP) BERT.

| Model | TT | TV | PE | SM | CP (ours) |
|---|---|---|---|---|---|
| Time | 0.005s | 9.226s | 0.011s | 1.283 | 0.121s |

Table 2: Time cost per query with 1K candidate images.

We adopt BERT-base model (Devlin et al. (2019)) as the backbone. It consists of 12 layers of Transformer blocks (Vaswani et al. (2017)). Each block has 768 hidden units and 12 self-attention heads. We load the weights of BERT-base pre-trained on text datasets as the initialization. The training is conducted on a Linux server equipped with 256GB memory and 4 V100 GPU cards. We use the ADAM optimizer. Following Li et al. (2020), 100 bounding boxes per image is extracted by faster R-CNN (Ren et al. (2015)) pre-trained on visual genome (Krishna et al. (2017)) provided by Anderson et al. (2018). We set the maximal query length as 44, which is the same as Li et al. (2020).

### 4.1 WITHOUT PRETRAINING

As shown in Table 1, text-vision (TV) BERT significantly outperforms two-tower (TT) BERT. In the query2image retrieval task, the recall@1 achieved by TT is only 58.3. In contrast, TV achieves a 66.3 recall@1. In the image2query retrieval task, TT only achieves a 68.3 recall@1. In contrast, TV achieves an 80.6 recall@1. The higher accuracy of TV validates the effectiveness of exploiting multi-modal attention. As shown in Table 1, the proposed cross-probe (CP) BERT also significantly outperforms TT. Meanwhile, it achieves a comparable recall@1 in query2image retrieval compared with TV. An interesting observation in Table 1 is that our CP outperforms SM. Note that, we set $l = 2$ and $L = 12$ for both CP and SM. We further compare the time cost. The experiments are conducted on a single V100 GPU. As shown in Table 2, TV is significantly slower than TT. Meanwhile, Poly-Encoder (PE) is much faster than TV, and is slower than TT. SM is faster than TV, but much slower than TT. Our CP achieves $77\times$ speedup over TV, and $10\times$ speedup over SM.

The proposed cross-probe BERT relies on $m$ text probes and $n$ vision probes. It adopts two-tower architecture in the lower $L - l$ layers and uses the text-vision architecture in the upper $l$ layers. Different from split-merge encoder, in the upper $l$ layers, only vision probes and text probes are involved in computing cross-modal attentions. Since the number of vision and text probes $(m, n)$ are much smaller than the number of words and bounding boxes $(M, N)$, the proposed cross-probe BERT is much more efficient compared with split-merge BERT when using the same $l$ and $L$.

**Influence of the number of probes.** We fix $l = 2$ and $L - l = 10$. As shown in Table 3, as $(m, n)$ changes from $(5, 5)$ to $(15, 15)$, the performance significantly improves. For instance, when $m = 5$ and $n = 5$, it only achieves a 62.2 recall@1 in the query-to-image retrieval. In contrast, when $m = 15$ and $n = 15$, it achieves a 67.0 recall@1. Nevertheless, larger $m$ and $n$ will lead to larger computation cost. Considering both effectiveness and efficiency, we choose $m = 5$ and $n = 15$.

| $m$ | $n$ | query2image R@ | | | image2query R@ | | | time per query |
| | | 1 | 5 | 10 | 1 | 5 | 10 | |
|---|---|---|---|---|---|---|---|---|
| 5 | 5 | 62.2 | 89.3 | 94.6 | 75.3 | 95.7 | 97.8 | 0.542s |
| 10 | 10 | 63.1 | 89.8 | 94.6 | 76.9 | 95.4 | 97.8 | 0.126s |
| 15 | 15 | 67.0 | 91.2 | 95.5 | 80.7 | 96.3 | 99.1 | 0.191s |
| 5 | 15 | 66.0 | 91.0 | 95.3 | 80.2 | 96.1 | 98.2 | 0.121s |

Table 3: The influence of the number of text and vision probes $m, n$. Experiments are on MS-COCO.

| $l$ | $L-l$ | query2image R@ | | | image2query R@ | | | time per query |
|---|---|---|---|---|---|---|---|---|
| | | 1 | 5 | 10 | 1 | 5 | 10 | |
| 4 | 8 | 65.8 | 91.2 | 95.5 | 80.7 | 96.3 | 98.4 | 0.250s |
| 2 | 10 | 66.0 | 91.0 | 95.3 | 80.2 | 96.1 | 98.2 | 0.121s |
| 1 | 11 | 63.6 | 89.9 | 94.7 | 75.3 | 96.2 | 98.4 | 0.063s |

Table 4: The influence of the number of text-vision layers $l$. Experiments are on MS-COCO.

| | MS-COCO | | | | | | Flickr30K | | | | | |
|---|---|---|---|---|---|---|---|---|---|---|---|---|
| | query2image R@ | | | image2query R@ | | | query2image R@ | | | image2query R@ | | |
| | 1 | 5 | 10 | 1 | 5 | 10 | 1 | 5 | 10 | 1 | 5 | 10 |
| TT | 62.7 | 89.3 | 93.5 | 76.2 | 95.9 | 98.6 | 57.7 | 81.2 | 86.7 | 73.4 | 92.6 | 96.0 |
| PE | 61.4 | 89.1 | 93.9 | 75.9 | 94.8 | 98.2 | 58.3 | 83.1 | 89.1 | 72.8 | 92.2 | 96.2 |
| SM | 62.4 | 90.1 | 95.6 | 75.3 | 94.8 | 97.8 | 58.6 | 83.3 | 89.8 | 74.2 | 93.1 | 96.4 |
| TV | 69.6 | 93.1 | 97.2 | 83.4 | 97.2 | 99.1 | 69.0 | 90.4 | 94.5 | 81.5 | 95.9 | 98.3 |
| CP | 70.9 | 92.5 | 96.6 | 83.3 | 96.9 | 99.4 | 69.1 | 89.8 | 94.1 | 83.5 | 96.0 | 98.0 |
| Uni-VL | 69.7 | 93.5 | 97.2 | 84.3 | 97.3 | 99.3 | 71.5 | 90.9 | 94.9 | 86.2 | 96.3 | 99.0 |
| UNITER | – | – | – | – | – | – | 72.5 | 92.4 | 96.1 | 85.9 | 97.1 | 98.8 |

Table 5: Comparisons among two-tower BERT, text-vision BERT, Poly-Encoder, and cross-probe BERT, Unicoder-VL and UNITER with pre-training.

**Influence of $l$.** We fix $L = 12$, $m = 5$ and $n = 15$. Since $L$ is fixed, the larger $l$ leads to a smaller $L - l$. As shown in Table 4, when $l = 1$, that is, using only a single cross-attention layer, our cross-probe (CP) BERT achieves a 63.6 recall@1 in query2image retrieval. It is significantly better than two-tower BERT with a 58.3 recall@1. In the meanwhile, when $l$ increases from 1 to 2, the recall@1 considerably increases to 66.0, whereas the text-vision (TV) BERT achieves a 66.3 recall@1. That is, using only 2 cross-attention layers in CP BERT has achieved a comparable performance with text-vision BERT with 12 cross-attention layers. Meanwhile, our CP BERT uses only 5 text probes and 15 vision probes, whereas TV BERT uses 44 word features and 100 bounding box features. When $l$ increases from 2 to 4, accuracy does not change considerably, thus we set $l = 2$ by default.

### 4.2 WITH PRE-TRAINING

Existing BERT models rely on pre-training on a large-scale dataset. We further evaluate the influence of pre-training on the proposed cross-probe BERT model. Following (Li et al. (2020)), we use two datasets, SBU Captions (Ordonez et al. (2011)) and Conceptual Captions (Sharma et al. (2018)), for pre-training. Conceptual Captions contains 3.3M image-caption pairs crawled from the web. Due to broken URLs, the number of image-caption pairs of Conceptual Captions dataset is around 3M. SBU Captions contains 1M image-caption pairs. Due to broken URLs, the number of image-caption pairs of SBU Captions dataset is around 0.8M. We pre-train all ranking models by triplet loss and the retrieval performance is shown in Table 5. As shown in the table, our cross-probe BERT achieves a comparable retrieval accuracy with text-vision BERT. Meanwhile, our CP considerably outperforms TT, PE, and SM. We also compare two recent text-vision BERT methods, Unicoder-VL (Li et al. (2020)) and UNITER (Chen et al. (2020)). Note that, Unicoder-VL is pre-trained on Conceptual Captions and SBU datasets as ours but UNITER is pre-trained on Visual Genome Krishna et al. (2017) and MSCOCO Lin et al. (2014) datasets in addition to Conceptual Captions and SBU datasets. As shown in Table 5, our CP achieves comparable performance with Unicoder-VL on MS-COCO dataset. But our performance is worse than Unicoder-VL (Li et al. (2020)) and UNITER (Chen et al. (2020)) on Flickr30K dataset. Note that, both Unicoder-VL and UNITER adopt text-vision BERT architecture, and thus they are significantly slower than ours.

## 5 CONCLUSION

Benefited from exploiting cross-modal attentions, text-vision BERT has achieved excellent performance in vision-language retrieval. Nevertheless, the extremely expensive computational cost of text-vision BERT limits its usefulness in large-scale search. To boost efficiency and meanwhile maintain the high effectiveness, we propose a novel ranking model, cross-probe BERT in this work. By utilizing devised probes, the cross-model attentions are conducted on a small number of probes, which is much more efficient than text-vision BERT. Systematic experiments conducted on two public datasets demonstrate the excellent effectiveness and efficiency of our cross-probe BERT.

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

# A    APPENDIX

## A.1    ABLATION STUDY ON SPLIT-MERGE BERT.

In the lower $L - l$ layers, split-merge BERT conducts two-tower architecture, which individually models the text feature and the image feature. In the upper $l$ layers, it adopts the text-vision architecture, which exploits the cross-modal attentions. When $l = 0$, split-merge BERT degenerates to two-tower BERT, which is fast but not effective. On the other hand, when $l = L$, split-merge BERT degenerates to text-vision BERT, which is slow but effective. We evaluate the effectiveness of split-merge BERT when $l \in \{0, 2, 8, 9, 10, 12\}$. We fix $L = 12$ and thus $L - l \in \{0, 2, 3, 4, 10, 12\}$ in this case. We conduct experiments on MS-COCO dataset. As shown in Table 6, when $l = 2$, the accuracy is slightly better than the two-tower BERT ($l = 0$), but much worse than the text-vision BERT ($l = 12$). When $l$ increases to 9, the query2image retrieval has achieved a comparable accuracy with text-vision BERT. But the time cost when $l = 9$ is also large, limiting its efficiency.

| $l$ | $L - l$ | query2image R@ | | | image2query R@ | | |
|---|---|---|---|---|---|---|---|
| | | 1 | 5 | 10 | 1 | 5 | 10 |
| 12 | 0 | 66.3 | 91.7 | 96.3 | 80.6 | 96.4 | 98.7 |
| 10 | 2 | 67.2 | 92.2 | 96.5 | 79.6 | 97.1 | 98.8 |
| 9 | 3 | 66.1 | 91.9 | 96.5 | 78.2 | 96.2 | 98.4 |
| 8 | 4 | 64.4 | 91.1 | 96.1 | 77.8 | 95.8 | 98.4 |
| 2 | 10 | 60.6 | 89.6 | 95.3 | 74.2 | 94.4 | 97.6 |
| 0 | 12 | 58.3 | 86.5 | 91.8 | 68.3 | 93.5 | 97.4 |

Table 6: The influence of the number of text-vision layers $l$ and that of two-tower layers $L - l$.

## A.2    ABLATION STUDY ON POLY-ENCODER.

It utilizes $m$ context codes to generate the $m$ query-context features. Then the query-context features are attended by the image global feature to generate the image-context query global feature. The similarity score between the query sentence and the reference image is determined by the cosine similarity between the image-context query global feature and the image global feature. Since the cross-modal attentions are only exploited in the last layer, it might not be able to effectively model the interactions between visual features and word features. We evaluate the performance of poly-encoder in cross-modal retrieval and vary $m$ among $\{5, 10, 20, 40\}$. The experiments are conducted on MS-COCO dataset. As shown in Table 7, the accuracy of poly-encoder increases as the number of context codes $m$ increases. This is expected since more context codes can encode richer information. Nevertheless, when $m$ increases from 20 to 40, the retrieval accuracy slightly decreases. The worse performance might be due to over-fitting. In the meanwhile, compared with the two-tower encoder, the performance of poly-encoder is better, validating the effectiveness of the cross-modal attention in the last layer. On the other hand, compared with text-vision encoder, the accuracy achieved by poly-encoder is much lower, validating our statement that the cross-modal attention in the last layer might not fully exploit the interactions between the query and the image.

|  | query2image R@ | | | image2query R@ | | |
|---|---|---|---|---|---|---|
|  | 1 | 5 | 10 | 1 | 5 | 10 |
| m=5 | 57.3 | 87.1 | 92.9 | 68.5 | 93.5 | 97.3 |
| m=10 | 58.7 | 87.7 | 93.4 | 70.2 | 93.7 | 97.8 |
| m=20 | **59.4** | **88.0** | **93.5** | **71.4** | **93.9** | **97.9** |
| m=40 | 58.6 | 87.7 | 92.9 | 70.6 | 93.4 | 97.7 |

Table 7: The influence of the number of context codes on the retrieval accuracy.

A.3 EXPERIMENTS ON MSCOCO5K

|  | MS-COCO | | | | | |
|---|---|---|---|---|---|---|
|  | query2image R@ | | | image2query R@ | | |
|  | 1 | 5 | 10 | 1 | 5 | 10 |
| TT | 39.3 | 70.2 | 80.2 | 54.8 | 80.7 | 88.3 |
| PE | 39.7 | 70.1 | 80.5 | 55.6 | 81.5 | 89.5 |
| SM | 42.9 | 73.3 | 82.8 | 58.8 | 83.8 | 90.9 |
| TV | 46.5 | 75.6 | 85.0 | 62.6 | 86.8 | 92.6 |
| CP | 46.8 | 75.8 | 85.0 | 62.9 | 86.7 | 92.7 |
| Uni-VL | 46.7 | 76.0 | 85.3 | 62.3 | 87.1 | 92.8 |
| UNITER | 48.4 | 76.7 | 85.9 | 63.3 | 87.0 | 93.1 |

Table 8: Comparisons among two-tower BERT, text-vision BERT, Poly-Encoder, and cross-probe BERT, Unicoder-VL and UNITER with pre-training.

| Model | TT | TV | PE | SM | CP (ours) |
|---|---|---|---|---|---|
| Time | 0.005s | 48s | 0.03s | 7s | 0.6s |

Table 9: Time cost per query with 5K candidate images.

Table 8 compares the accuracy of the proposed CP methods and the baseline methods as well as existing state-of-the-art methods. As shown in Table 8, our CP method achieves a comparable accuracy with text-vision BERT model (TV), Unicoder-VL and UNITER.

Table 9 compares the time cost per query used in all methods. The experiments are conducted on a single V100 GPU. Note that, TV method takes more than 2 days to finish the testing on MSCOCO 5K split, which is extremely slow.

