# OpenReview forum: "Cross-Probe BERT for Efficient and Effective Cross-Modal Search"
_ICLR.cc/2021/Conference — Reject_

### Official Review · AnonReviewer1 · 2020-10-27
**Review comments for “Cross-Probe BERT for Efficient and Effective Cross-Modal Search”**

**Rating:** 6
**Confidence:** 4

**Review:**

The efficiency of current SOTA methods on image-text retrieval, especially those visual-language cross-modality pre-training models, has always been a critical problem compared with traditional joint-embedding models. This paper aims to solve this problem by integrating the efficiency of two-tower models in lower layers and the effectiveness of cross-attention learning models in higher layers. Specifically, this paper proposes to generate several vision and text probes with local features to decrease the cost of computation. Experiments on MSCOCO 1K test set and Flickr30K are conducted to show the effectiveness of the proposed method.

The motivation is insightful, and the method seems reasonable, while the experiment is weak. Following is my detailed concerns.
1.	The paper shows the overall cost of models while lacks the detailed analysis of each part in the model. For example, is the computational cost of each BERT layer on query and image the same? What is the unit cost for feed-forward and self-attention operation? These also have an influence on the total cost of the models.
2.	The operation cost on CPU is not shown in the paper.
3.	Since the key of this paper is to discuss the efficiency of retrieval model, the influence of different parameters (e.g., number of probes, l) in terms of efficiency is very important. However, only comparison of performance is shown in the paper.
4.	Result of performance and cost time on 5K test set would better demonstrate the efficiency of the proposed model, while this is lacked in this paper.
5.	This approach tries to improve efficiency of image-text retrieval while has some sacrifice on effectiveness. While most SOTA methods on this problem is based on text-vision (TV) BERT. How to effectively adapt this approach to existing models and the generalization of this method is not discussed in the paper.

Overall, I think this paper could be improved, especially in experiment part.

---

### Official Review · AnonReviewer2 · 2020-10-27
**This paper presents a method dubbed cross-probe BERT for cross-modal retrieval，which relies on devised text and vision probes, and the cross-modal attention is conducted on text and vision probes. Thus the cross-modal attention can be exploited with a lightweight computation cost in this method. Experiments demonstrate its effectiveness and efficiency.**

**Rating:** 6
**Confidence:** 4

**Review:**

This paper develops a architecture termed cross-probe BERT to deal with the costly computation from cross-modal attentions used in search task. Motivated by the success of the “split-merge” style encoder in query document retrieval, it extends text-vision BERT to adopt it for speeding up the computation. In particular, it devises several vision probes and text probes along with the image’s local features and the query’s word features. Besides, the cross-modal attention is conducted on text and vision probes. Therefore, it can exploit cross-modal attention with a lightweight computation cost. Systematic experiments demonstrate the state-of-the-art effectiveness and efficiency of the proposed method.

Pros:
- Relative strong results compared to baselines about the effectiveness and efficiency of cross-modal retrieval.
- Paper is well written and easy to follow.

Cons:
- Limited contribution of the proposed method. This paper extends text-vision BERT to adopt it for speeding up the computation. Besides, ideas are similar to the combination of split-merge BERT + Poly-Encoder based on the architecture in Figs 2 and 3.
- The hyper-parameters \alpha and \beta in the equation of complexity of transformer layer are not clearly stated. What’s the effect and role about them? In addition, the corresponding analysis is not provided.
- Insufficient experimental analysis, such as the influence of the number of probes.

---

### Official Review · AnonReviewer4 · 2020-10-27

**Rating:** 5
**Confidence:** 5

**Review:**

Summary:
1. This paper proposed a new architecture to accelerate the Cross-Modal BERT inference speed. The motivation is current Cross-Modal BERT is too slow during the online inference, especially for the large data. The bottleneck is due to the cross-modal attention. This paper proposed a new architecture that accelerate the inference speed, while keeps the retrieval performance.

Strength:
1. The motivation is clean. The slow inference speed problem of current image-text BERT is an important problem.
2. The idea is clean, although there might be some mistakes (listed in the weakness section). The idea basically contains two parts. First, caching as much information as possible during inference. Therefore, the proposed approach applies less layers for Cross-Modal attentions. Those cross-modal attention layers are applied at the end of the architecture. Second, accelerate the cross-modal attention by using less tokens for this module.
3. The performance is good when comparing with the SoTA w/ and w/o pre-training.

Weakness:
1. The method seems not novel.  The method seems like a merge of Split-merge model and the Poly-Encoder model.
2. The equation for $\alpha_{CP}$ might not be correct. As $\alpha_{CP}$ is the speed-up ratio of the proposed cross-probe BERT over the text-vision BERT, it should be $\alpha_{CP} = \frac{\hat C_{TV}}{\hat C_{CP}}$. Furthermore, the numerator seems incorrect. If the numerator is for $\hat C_{TV}$, it should not include $C_T$.
3. The Table 3 is a little bit surprising. The performance of $m = n = 15$ is even higher than the performance of cross-modal BERT (TV in Table 1). As the cross-modal BERT has much more cross-attention layer than the proposed approach, I somehow feel the TV model is not trained properly. The lower performance of TV might indicate the TV model is overfitting.
4. For Table 4, the CP model achieves higher (or similar) performance than Unicoder-VL on COCO, but worse performance on Flickr30k. I wonder what is the reason?
5. If I remembered correctly, ViLBERT is pretrained using only Conceptual-Caption data (SBU is not used), while the CP is pretrained using both SBU and Conceptual-Caption. Therefore claiming higher performance than ViLBERT might be misleading.
6. The Table 5 is incomplete. More recent works should be included and compared in the Table, for example UNITER: Learning UNiversal Image-TExt Representations.
7. The image-text BERT model is very sensitive to the hard negative mining during training. UNITER and UNICODER-VL uses online hard negative mining, which would leads to better performance, while ViLBERT uses offline hard negatives. I wonder what is the implementation detail for Table 1 and Table 5? Specifically, are the model in Table 1 trained using the same hard negative mining strategy?
8. The paper shows the inference speed at 1K testing case. As COCO has 5K testing split, I wonder what is the speed and image retrieval accuracy for 5K testing case? What is the speed-up ratio against TT, PE, and SM over 5K split? Showing speed-up ratio in large amount of data is important for judging the effectiveness of the proposed approach.
9. Why PE is much faster than the proposed approach (CP) in Table 2?
10. As in the method part, the author shows the theoretical speed up ratio. I wonder what is actual number of theoretical speed-up for CP over TT, PE, SM, and TV.

---

### Official Review · AnonReviewer3 · 2020-10-31
**Official Blind Review #3**

**Rating:** 6
**Confidence:** 3

**Review:**

This paper proposes an improved method, Cross-Probes BERT,  based on the split-merge architecture to further accelerate cross-modal retrieval. Specifically, it devises several vision probes and text probes along with the image inputs and query inputs in the low-level two-tower architecture. After that, the attended vision and text probes are concatenated and fed into cross-modal self-attention layers to interact in the high-level one-tower architecture. The experiments on two benchmarks, MS-COCO and Flickr30K show quite promising performance, especially on computation efficiency.

Overall, the definition of the task and the overall architecture of this paper are both clear and straightforward, making this paper easy to understand. The idea of introducing some probes to collect information and reduce the computation cost in the split-merge architecture makes sense to me. Two kinds of probes, namely vision probes and text probes, are proposed to collect the global features of image inputs and query inputs, which is reasonable and interesting. Besides, the paper provides comprehensive experiments on performance and computation cost to validate the effectiveness and efficiency of their proposed method.


Here are some of my questions and concerns for the paper:
1. Compared with previous works (such as Ploy-Encoder, BERT_{SM}, DCBERT), the proposed method seems a bit incremental.
2. The architecture proposed in this paper seems to be applicable to text retrieval tasks. Why not test the model in some text-text retrieval tasks explored in Ploy-Encoder and DCBERT.
3. According to Section 3.5, the \overline{w} is the sum-pooling of \hat{w}_1, …, \hat{w}_N while it is the output of l transformer layers in Figure 3. Is there something wrong in Figure 3? Please make it clear.

---

### Decision · Program_Chairs · 2021-01-07
**Final Decision**

**Decision:**

Reject

**Comment:**

After the rebuttal phase, all reviewers give borderline scores (leaning slightly positive, one of these noted in the comment rather than final review). While the reviewers recognize the potential merit of the contribution (efficiency while preserving effectiveness), support for acceptance is not sufficient. The major concerns include novelty (shared by multiple reviewers) and the limited experimental settings.